# Influence of Effective Microorganisms, Colloidal Nanosilver and Silver Compounds on Water Content in New and Used Engine Oil: A Preliminary Study

**Rafał Krakowski**

Department of Marine Propulsion Plants, Faculty of Marine Engineering, Gdynia Maritime University, Morska Street 81-87, 81-225 Gdynia, Poland; r.krakowski@wm.umg.edu.pl

**Abstract:** This paper presents research upon the impact of ecological measures on the water content of new and used oil. Water and microorganisms are major factors of contamination for engine oils and fuels, and they lead to a significant reduction in the lifetime and performance of engines. The microorganisms occur naturally in the environment, from which they enter into the oil and fuel. Despite various preventive measures, it is not possible to completely remove water from petrochemical products. That is why the protection against and prevention of the various types of contamination of petroleum products, especially microbiological contamination, is very important. Biocides are one example of the agents used for the prevention of contamination; biocides belong to a group of pesticides that are used to eliminate microbial contamination. Due to the fact that currently available methods are ineffective and often have a detrimental effect on the natural environment, research is underway to discover modern and ecological measures to combat the phenomenon of the microbiological contamination of petroleum products. This paper shows the effects of environmentally friendly additives on the water content in lubricating oil, i.e., whether these additives cause the release of water bound in the oil, and whether their composition increases the water content of the oil. Samples of new and used oil were utilized for the tests. Effective microorganisms (EM), in both liquid form and within ceramic tubes, were added to the new and used oil samples. In addition, silver solution and silver compounds were added in the same amounts as the liquid effective microorganisms. In order to confirm and compare the obtained test results, the dynamic viscosity, flash point, acid number, base number, and water content were measured, and these are presented in this study. These measurements were followed by the analysis of the influence of the additives on the water content. It was found that the adding of effective microorganisms to fresh oil in liquid form caused a significant increase in the water content of the oil; in the case of a smaller amount (2.5 mL), the water content more than doubled, and in the case of a larger amount, the water content exceeded the measuring range of the device. Next, an evaluation of the impact of these environmental measures on the water content was carried out. It was found that the adding of liquid effective microorganisms to new oil caused a significant increase in the water content of the oil; the water content more than doubled for the addition of 2.5 mL, while for a larger addition (5 mL), the water content exceeded the measuring range of the device. The same was true for the addition of silver compounds, regardless of their type and amount. The best results were obtained with ceramic effective microorganisms, but the results for silver cannot be presented due to the over-titration of the device (the amount of water exceeded the measuring range). For used oil, the liquid effective microorganism was found to give the best result.

**Keywords:** engine oil; water content; oil properties; effective microorganisms; silver compounds



## 1. Introduction

The microbial contamination of both fuels and engine oils has been the subject of many past studies. At the same time, this problem is still valid and has become more and

more popular as an area of research. Studies have shown that microbial activity can be detrimental to the operation of various modes of transport [1–5]. It is important to prevent microbial contamination that can occur in petroleum products. Microbial contamination can be prevented through proper cleansing, decontamination, or by chemical methods. Another method of protection against microbial contamination is the use of specialized coatings on fuel tanks [6].

One means of combating harmful organisms are commonly used synthetic or natural compounds, i.e., biocides. Cleaning agents are also used during fuel storage to help reduce the amount of water in the tank. Currently, for the sake of environmental protection, it is necessary to develop new methods to prevent the growth of harmful microorganisms in engine oil. When developing methods of eliminating the phenomenon of the microbiological contamination of petrochemical products with the use of ecological measures, it is necessary to take into account both the structure and the phenomena occurring in the supply systems and the lubrication of machine elements [7–9].

Ecological methods of improving the performance parameters of engine oils may include the addition of effective microorganisms [10–14], i.e., a set of cultures of naturally occurring, non-genetically modified microorganisms; these are specially selected, microscopic organisms found on Earth. These microorganisms are used, among others, in industry, medicine, horticulture, and also in environmental protection. Another environmentally friendly measure involves the use of silver [15]. This element has long been used for protective and healing purposes. It is also applied in filters, and to clean the air of microorganisms and water in closed spaces, e.g., on airplanes. The method of silver ionization contributed to the enhancement of the disinfecting effect of this element. Silver ionization, as with copper ionization, has found application in the qualifying selection of some bacteria [16–20]. Silver is also used in the form of nanosilver. This article presents a study of the influence of additives commonly recognized as environmentally friendly on the water content in new and used oil.

## 2. Oil Contamination with Water

One of the tasks of engine oils is the reduction of friction between the surfaces of two moving and interacting elements of mechanical devices. Lubricating oils consist of a base oil and an active ingredient. The base oil gives typical characteristics to the lubricating oil, and the actual performance is determined by the active substances that improve the base oils in terms of resistance to oxidation, corrosion protection, wear protection, as well as wetting and emulsifying ability [21–31].

The problem is water bound in the lubricating oil. Water in oil comes in three forms: dissolved water, emulsified water, and free water. The point at which the oil cannot absorb more dissolved water is called the oil saturation point [32]. If more water enters the oil, the excess water will appear as separated water or an emulsion. Generally, if the saturation point is exceeded, the oil changes to a milky appearance. The water amount in the oil depends mainly on the additive package, operating temperature, pressure, and the quality of the base oil. For example, a highly refined oil with few additives will withstand a small amount of water before being saturated, around 100 ppm at 21 °C. On the other hand, an ester-based oil can withstand 3000 ppm of water at 21 °C and above [31–33]. Water contained in oil systems deteriorates the rheological properties of the working fluid used, reducing its lubricating and insulating abilities. Water in oil reduces the possibility of the transference of bearing loads, accelerates the processes of oil oxidation, rinses out improvers, increases the amount of deposits, and causes corrosion.

Water promotes the aeration of oil, which causes a foaming problem and has a negative effect on oil improvers, causing them to undergo the process of hydrolysis or be washed out [34–36].

## 3. Materials and Methodology

### 3.1. Materials Used in the Research

For the tests, 5W30 synthetic oil was used. It is ESP (emission system protection) formula fully synthetic oil manufactured by Mobil. This oil changes its viscosity slightly during temperature changes. As a result, it perfectly lubricates engine parts, while protecting against carbon deposits, sludge, and other harmful impurities that can accelerate wear.

The described oil retains its optimal viscosity from −30 °C and its fluidity at −35 °C; 5W30 oil has high pumpability, thanks to which it is highly effective in cleaning the most important engine components, while also contributing to their efficient cooling [37].

A 5W30 automotive oil, according to the SAE J300 standard of 2015, must meet the following criteria [38]:

- Maximum dynamic viscosity of 6600 cP at −30 °C;
- Maximum pumpability temperature of 60,000 cP at the temperature of −35 °C;
- Kinematic viscosity at temperature 100 °C min, 3.8 $mm^2/s$ to 9.3–12.5 $mm^2/s$;
- Viscosity HTHS in 150 °C min, 2.9 cP.

The used oil is of the same type as the new oil, and is sourced from a compression ignition internal combustion engine. Its mileage is approximately 1000 km, having been in operation for a year in various real conditions on mixed routes (city and motorway routes).

Effective microorganisms and silver were added to the oil about 4 weeks prior to beginning the research.

The effective microorganisms used comprised 81 different strains of aerobic and anaerobic microorganisms, including lactic bacteria (*Lactobacillus casei, Streptococcus lactis*), yeasts (*Saccharom–yces* spp., *Candida utilis*), photosynthetic bacteria (*Rhodopseudomonas palustrus, Rhodobacter spae*), molds (*Aspergillus oryzae, Mucom hiemalis*), and actinomycetes (*Streptomyces, Streptomyces griseus*) [39].

Photosynthetic bacteria under certain conditions, for example, $CO_2$ or property temperature, produce useful biochemical active substances from organic matter or toxic gases. Lactic acid bacteria also slow down the growth of harmful bacteria. Another important component of the mixture is yeast. Mushroom fermentation breaks down organic matter and neutralizes unpleasant odors.

The action of EMs includes natural processes, and furthermore, as these microorganisms are not genetically modified, they are therefore environmentally friendly [40]. Effective microorganisms in the form of ceramic tubes are a special type of clay inoculated with beneficial microflora (fermented clay), which mature in natural conditions for several months, before being fired under high temperatures in anaerobic conditions; they include compositions of beneficial microorganisms, sugar cane molasses, and restructured water. Clays inoculated with microbes mature for many weeks in appropriate humidity and temperature levels, and then, depending on their intended use, they are fired at high temperatures (gray) of 1200–1300 °C. When fired, they become a viscous liquid, and when cooled to ambient temperature, they bind strongly and are extremely hard. Ceramics fired using this method are less porous, denser, and less brittle. The raw materials used in traditional ceramics include minerals such as clay, silica stone, and feldspar. Ceramics are inorganic and contain no metals. The ceramic tubes used were shaped and then hardened by firing at high temperatures. Ceramics do not react chemically, do not burn, and are harder than steel. The shapes of the fired elements or dishes are adjusted in accordance to their future use, and the content of effective microorganisms is 3% in each ceramic tube [41].

Effective microorganisms in the commercial forms presented in Figure 1 were used for the research with the following composition: water: 94%; Effective Microorganisms® (lactic acid bacteria, photosynthetic bacteria, saprophytic bacteria, yeast, fungi, and actinomycetes): 3%; molasses: 3%.

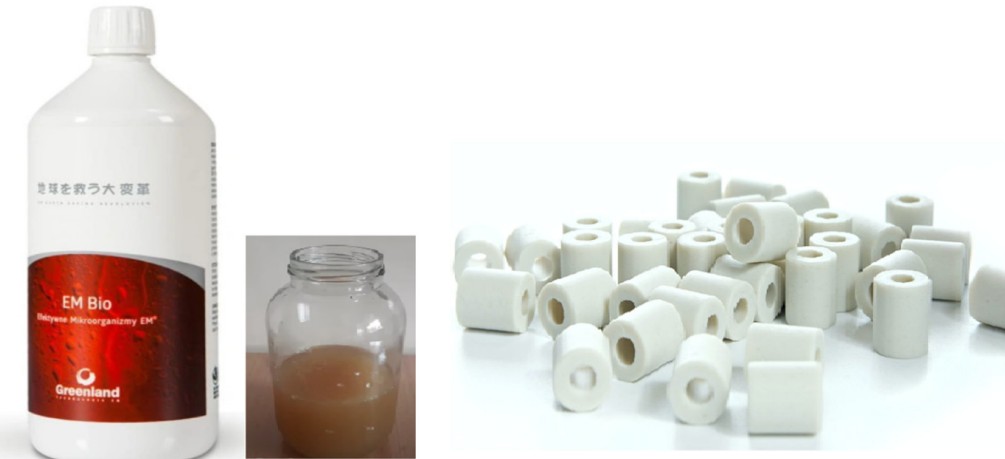

**Figure 1.** The commercial forms of effective microorganisms [42,43].

An EM product manufactured by Greenland Technologia was used in the research. The company deals with the production and sale of products based on the technology of Effective Microorganisms®, which are widely used in agriculture and horticulture, animal husbandry, or in biological environmental treatment (municipal and home sewage treatment plants, regeneration of soil and water reservoirs).

Greenland Technologia EM Ltd. is the sole licensee of EM Research Organization Inc. (EMRO) for Poland and Central and Eastern Europe. The Japanese company EMRO is also a licensor for producers and distributors of products in several dozen countries on all continents.

The second additive used in the research is colloidal silver, also known as nanosilver and silver solution.

Ion silver is as transparent as water, is stored in glass bottles, and consists of 90% silver ions and only about 10% silver particles. This type of silver is called "silver solution". After entering the body, silver ions immediately combine with it, creating an insoluble salt–silver chloride. Such a property is not exhibited by electrically neutral silver particles. This salt does not have any properties that are attributed to colloidal silver.

Nanosilver consists of microscopic particles and silver ions that can only be observed through an electron microscope. Due to the high concentration of silver nanoparticles, colloidal silver does not resemble water. Its characteristic feature is a clear color, which can range from yellow to amber. Colloidal silver has a darker color because the silver particles dispersed in the water block the light passing through them with a specific wavelength, which is approximately 400 nm. Additionally, colloidal silver can be stored in plastic bottles because silver nanoparticles are chemically insensitive and completely resistant to the damaging effects of UV radiation.

Colloidal silver with silver nanoparticles is the least common type of silver commercially available due to its complicated production process and high production costs. However, it is the only effective silver with scientifically proven properties. Real colloidal silver consists of 80% silver particles, with the remaining 20% made up of silver ions. The decisive factors for the quality of real colloidal silver are the content of silver particles and their very active total surface. In colloidal silver, nanometer-sized particles form a colloid, so it is not necessary to use a protein substrate to suspend them in water. Due to their electrokinetic potential, nanoparticles repel each other, keeping the colloid stable.

Silver suppressed to the form of nanoparticles has a large active surface, and enormous biocidal potential. The productive value of nanosilver includes the eradication of over 99.99% of bacteria, viruses, mold, and fungi. Good carriers for nanosilver are activated carbon fibers (ACFs), commonly used in sewage treatment plants for removing various types of contamination [44].

Experimental data from studies [45] show that silver nanoparticles at a concentration of 4 mg/mL completely inhibit the growth of *Staphylococcus aureus* bacteria. The results of transmission electron microscopy confirmed the damage to the cell walls caused by silver nanoparticles and the accumulation of these particles in the bacterial cell membrane [46,47].

Silver solution and colloidal nanosilver was used for the tests in the commercial form shown in Figure 2. Silver solution (transparent), sold as Silver Klar Ag+, was created on the basis of demineralized water and ionic silver (25 ppm), and was produced by the Naoliwieni company located in Rosanów (Poland). Colloidal silver (amber), sold as ARGENTUM200, was created on the basis of demineralized water with a colloidal non-ionic silver content of 25 ppm, and was manufactured by Aura Herbals Ltd in Sopot (Poland).

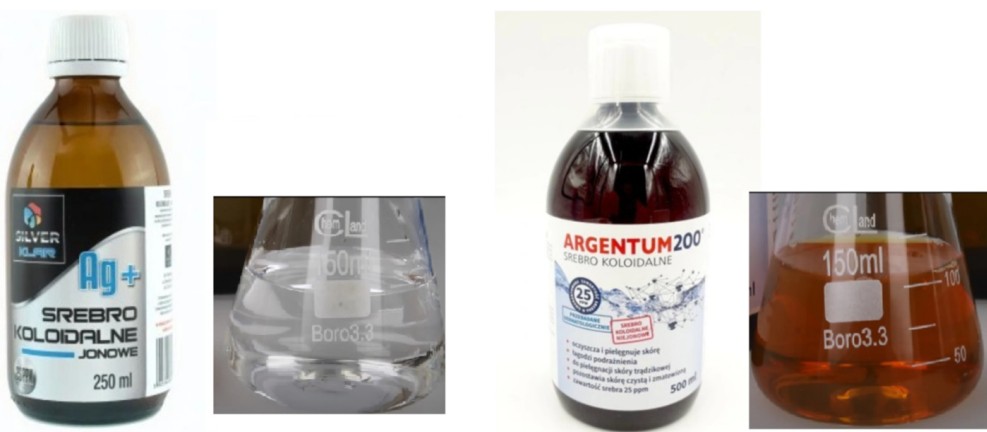

**Figure 2.** The commercial forms of colloidal nanosilver and silver solution [48,49].

### 3.2. Methodology

The water content of new and used oil with and without any additives was tested using the Cou-Lo AquaMax KF automatic titrator for water content determination in accordance with the Karl Fischer method. This device (Figure 3) uses the coulometric titration technique to ensure the repeatable and reliable measurement of the water content from 1 ppm upwards. It uses the technique of the precise generation of iodine into the system as a result of electrolysis, taking place in a very tight measuring vessel with the lowest level of moisture penetration from the outside, i.e., the so-called "drift". The selected parameters of the Cou-Lo AquaMax KF automatic titrator are shown in Table 1.

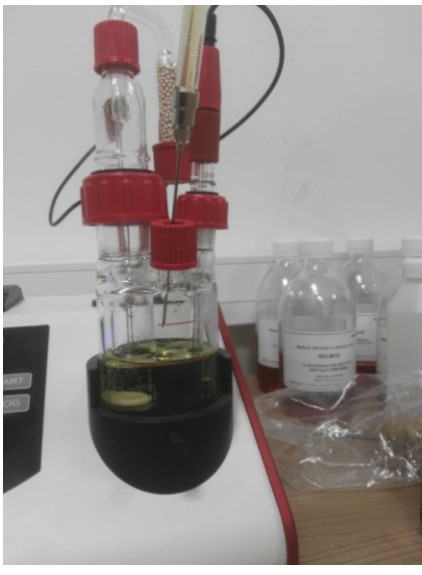

**Figure 3.** Tested oil with solvent and iodine on Automatic Cou-Lo AquaMax KF titrator.

**Table 1.** Selected parameters of the Cou-Lo AquaMax KF automatic titrator.

| Parameters | Cou-Lo Aquamax KF Moisture Meter |
|---|---|
| Titration method | Coulometric Karl Fischer titration |
| Endpoint detection | AC polarity |
| Measurement range | 1 μg–10 mg of water, max 200 mg of water |
| Moisture range | 1 ppm–100% water |
| Maximum titration rate | 2.24 mg per minute |
| Maximum electrolysis current | 400 mA |
| Drift Compensation | Controlled automatically |
| Minimum titration time | 0–30 min, user programmable |
| Precision | 10–100 μg ± 3 μg, 100 μg–1 mg ± 3 μg (ppm), over 1 mg ± 0.3% |
| Display format | μg, mg/kg, ppm, % |

New and used oils with liquid and ceramic effective microorganisms, silver solution, and colloidal nanosilver, were used for the research, wherein 2.5 mL and 5 mL of additives per 100 mL of oil, and three or six pieces of ceramic tube with diameters of 9 mm and heights of 11 mm were added to the samples.

In order to obtain the results of the water content of the oil itself and of the oil with additives:

1. The Aquagent Coulometric solvent (100 mL) was poured into the measuring vessel;
2. An ampoule of iodine in the amount of 5 mL was added;
3. The parts were twisted together and the device was placed on the stand;
4. The "G" generation electrode and the "D" detection electrode were connected;
5. It was then possible to start conditioning, i.e., to begin removing the moisture from the measuring system in order for the obtained measurement to correspond to the tested sample;
6. It was necessary to fill the glass syringe with test oil and weigh the oil-filled syringe, then the sample itself, to obtain the weight of the test sample;
7. After completing the measurement of the water content, the previously measured mass of the tested oil was fed into the device in order to obtain the water content as a percentage.

There is a platinum charge-generating electrode in the measurement system, which is responsible for the release of stoichiometric amounts of iodine into the system (with a ceramic diaphragm), precisely, optimally, and according to the following relationship: the 10.71 coulomb charge generated in the system releases the amount of iodine that balances the content of 1 mg of water. The AquaMAX KF device works according to the rules of coulometry and determines the water content of the sample by coulometric titration until the end point is reached. The end point means that free iodine appears in the system. Stoichiometric: 1 mole of water reacts with 1 mole of iodine. This means that 1 mg of water (0.001 g) is equivalent to 10.71 electric coulombs (1C = 1A × 1s). The device determines the water content in the tested sample by measuring the total amount of charges generated as the sum of the electrolysis current needed to produce the required amount of iodine in the system to bind the water contained in the tested material [50].

## 4. Results and Discussion

The research results are presented in the form of graphs in Figures below. These charts demonstrate the analysis of the influence of each of the additives on the amount of water in the oil samples. Each sample of pure oil and oil with additives was tested three times. However, this article presents the average result for each case.

Analyzing the obtained outcomes, it can be noted that in Figure 4, the water content of fresh oil with no EMs is 0.096%. The addition of 2.5 mL of microorganisms to the oil causes a considerable increase in the amount of water in the oil, because it is more than twice as high compared to oil without such an additive and amounts to 0.21% of the water content. With the addition of more microorganism solution, i.e., 5 mL, it was not possible

to determine the water content due to over-titration. It follows that this amount of water in oil was above 1%, i.e., above the measuring range of the device.

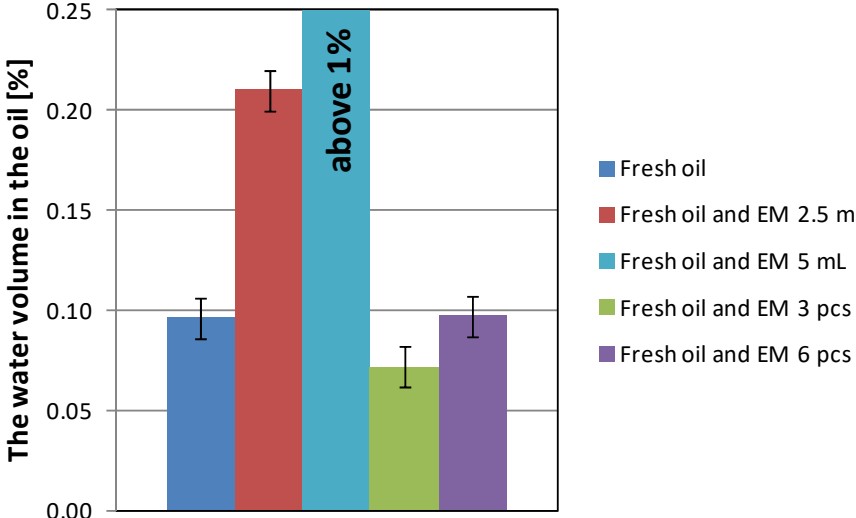

**Figure 4.** Water content in fresh oil and fresh oil with EMs in liquid form and ceramic tube form.

The situation was different with the addition of ceramic tubes, wherein the microorganisms are bound and do not mix with the oil as in their liquid form. In Figure 4, we can see that the addition of microorganisms, especially in a smaller amount (3 pcs), reduced the water content in the oil, which is positive from an oil exploitation point of view, while the greater number of tubes does not affect the water content of the oil in any significant way.

Figure 5 shows the color of the samples, and demonstrates that after adding microorganisms in a liquid form to the new oil, they became an oil–water emulsion and in this form were not suitable for use as a lubricating oil (there is probably some decomposition and release of water bound in the oil or a chemical reaction that resulted in the formation of such a large amount of water, which is related to the composition of the liquid additive of effective microorganisms).

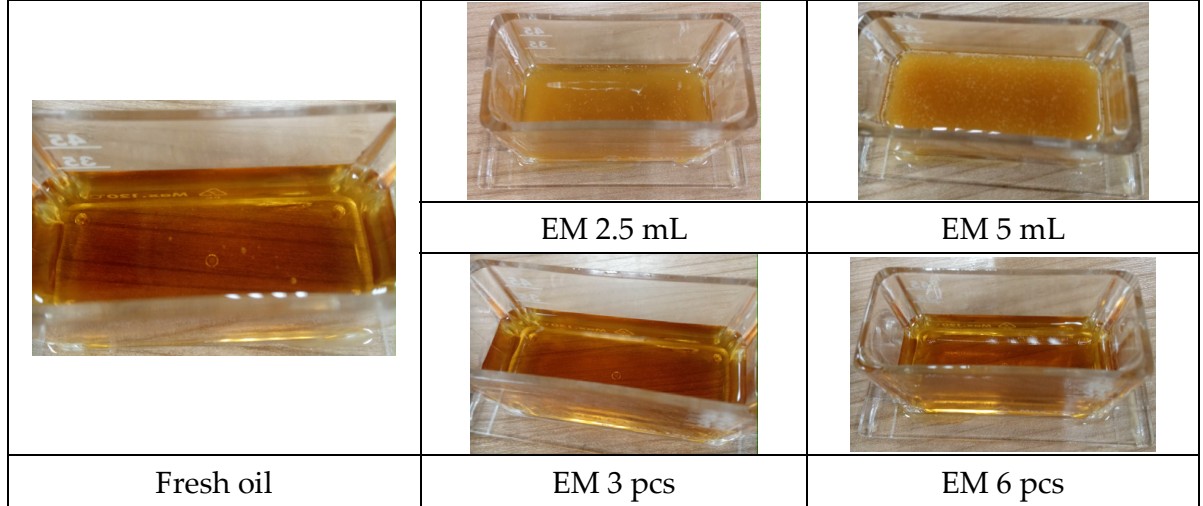

**Figure 5.** Coloring of fresh oil, with the addition of effective microorganisms in liquid form and ceramic tube form.

Silver compounds were also added to the new oil in amounts of 2.5 mL and 5 mL. In this case, it was not possible to read the results of the water content in the oil with the addition of both the first and the second type of silver for each amount, because after

prolonged operation, the titrator showed over-titration, i.e., the water content in the oil exceeded the measuring range of the device. The mixtures of oil with the addition of silver solution and colloidal nanosilver are presented in Figure 6.

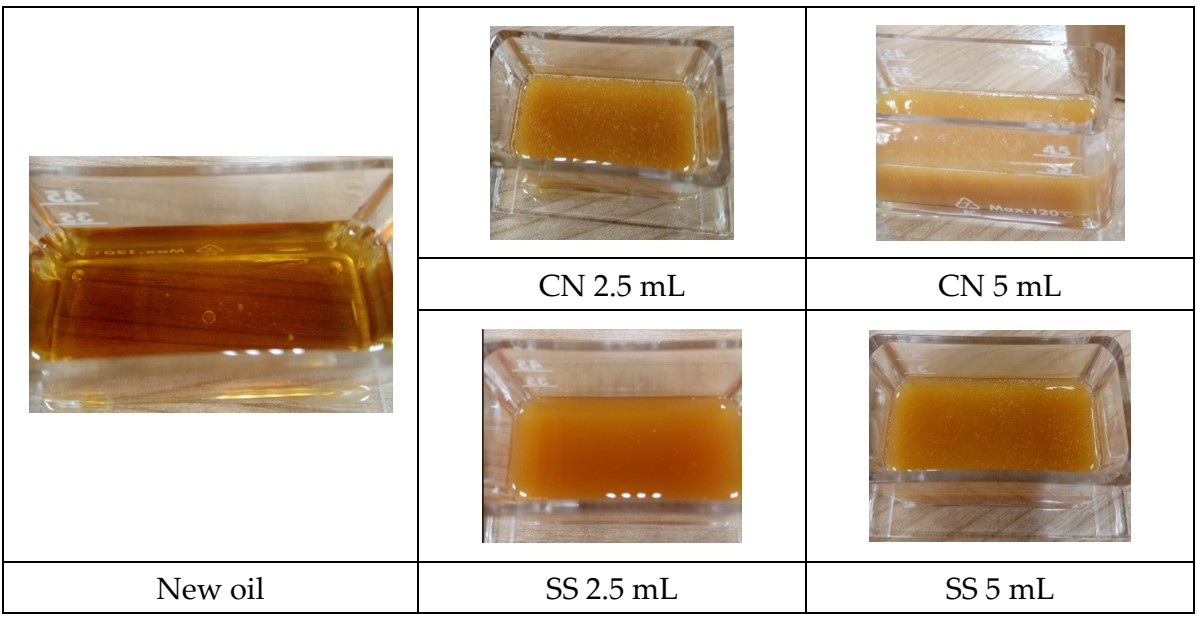

| | |
|---|---|
| CN 2.5 mL | CN 5 mL |
| SS 2.5 mL | SS 5 mL |

New oil

**Figure 6.** Coloring of new oil, with the addition of silver solution (SS) and colloidal nanosilver (CN) in liquid form.

These photos show that additives caused the formation of a water–oil emulsion. In this form, the lubricating oil will not properly perform the tasks for which it was intended. The milky appearance may be the outcome of Pickering emulsification [51–53], in which the bacteria and colloidal silver accumulate at the water–oil interface and prevent the water-in-oil droplets from coalescing. In the case of EMs, emulsification may also be caused by biosurfactants [54].

Figure 7 shows the flash point of fresh oil with the same additives as for the water content test. As in the case of the water content results, the best value of the flash point was obtained for three picks of ceramic-based effective microorganisms. The flash point was similar to new oil without additives. The remaining samples achieve results in a similar range of approximately 200 °C, which disqualifies them; due to safety reasons, the flash point should have the highest possible value.

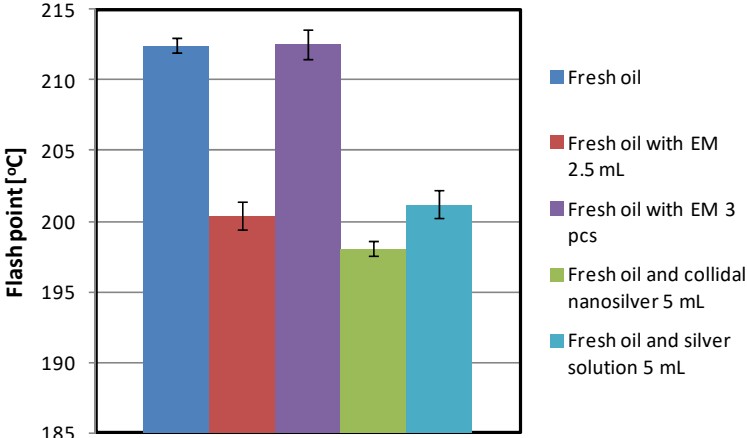

**Figure 7.** Flash point for oil with effective microorganisms (EM), colloidal nanosilver (CN), and silver solution (SS).

Statistical analysis was performed for a significance level of 95%. The obtained results for the water content and the flash point for the new oil and the new oil with additives for this significance level do not show any significant errors, because they fall within the lower and upper confidence intervals.

In order to confirm the obtained results, the article also presents a graph of viscosity versus temperature (Figure 8). In this diagram we can see that the curve of viscosity versus temperature for oil with the ceramic tubes is very similar to the curve for new oil without additives. On the other hand, the highest viscosity, which differs significantly from the viscosity of the oil without any additives, is of the oil with 2.5 mL of EM. Viscosity is 720 mPa·s at approximately 2 °C, which is similar to that of used oil 6, for which the viscosity at this temperature is 719 mPa·s. Other additives, including silver compounds, also increase the viscosity of the oil, but not as much as liquid effective microorganisms.

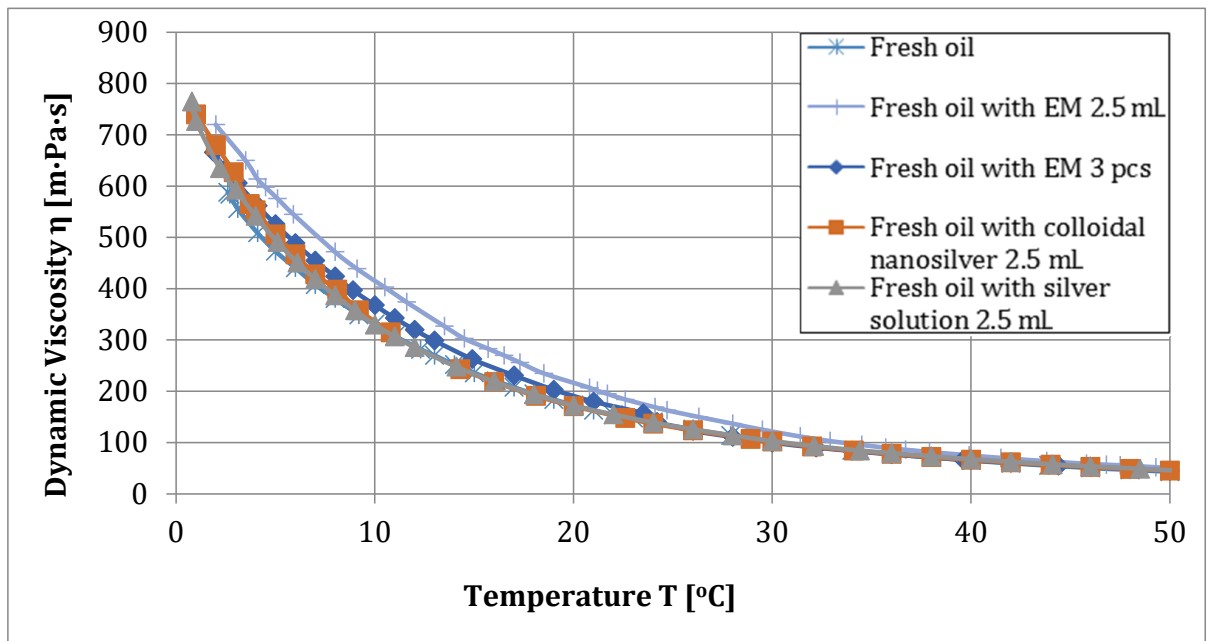

**Figure 8.** Viscosity for oil samples with the addition of effective microorganisms (EM), colloidal nanosilver (CN), and silver solution (SS).

Although effective microorganisms contain molasses, which is rich in sugar, no adverse effects on the rheology of the engine oil have been observed. At this stage, the dynamic viscosity was measured with a vibration viscometer. The obtained results demonstrate that the addition of ceramic effective microorganisms is more effective in new oil, whereas their liquid form is more effective in used oil. This should be confirmed by future research, and the effect of additives on the elemental composition of the oil and investigation into whether rheological shear thinning is evident are other potential areas of research. These tests will check whether molasses has a negative effect on the performance of the engine oil.

Figure 9 shows the water content in used oil (additives were added after using the oil). The chart compares the water content of oil without additives with that of oil that has been mixed with effective microorganisms and silver compounds. For a better illustration, the graph also shows the result for new oil. From this figure, we can see that for used oil the water content is approximately 0.17%, while for liquid microorganisms in a quantity of 2.5 mL, this amount is 0.6%, and for 5 mL, the result was not obtained due to over-titration, as was the case with new oil. This means that the water content was above the measuring range. As for new oil, the situation was different for microorganisms in a ceramic tube form. The water content for three ceramic tubes was about 0.02%, for six tubes it was 0.05%, while for new oil the water content was 0.09%. The addition of silver, regardless its type and

amount, is also not possible to present on the graph due to over-titration, i.e., the amount of water with these additives was above 1%.

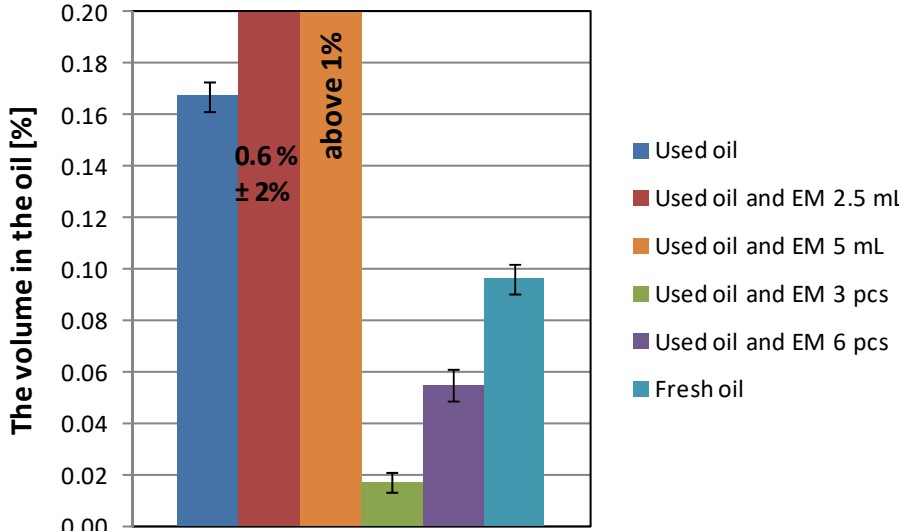

**Figure 9.** Water content in used oil and used oil with effective organisms in liquid form and ceramic tubes.

Similarly to new oil, in this case the lowest water content was obtained with the addition of EMs in the form of ceramic tubes, in a smaller quantity. Previous research [55–57] also shows that better parameters for used oil are obtained by adding EMs in liquid form. An example is the flash point shown in Figure 9. Moreover, Tables 2–5 show a comparison between the parameters of fresh and used oil without additives, and those in which the additives of effective microorganisms and silver compounds were added. Furthermore, analyzing the data from the tables, it can be concluded that the best additive for new oil is the ceramic tube form of effective microorganisms, and for waste oil, the addition of liquid effective microorganisms works best. As for the silver solutions, they adversely affect the parameters of both new and used oil. All the presented results confirm the obtained results, which show that microorganisms in the shape of ceramic tubes fulfill their role better in new oil, compared to in liquid form in used oil. Colloidal silver deteriorates the properties of both new and used oil, regardless of its type.

**Table 2.** Comparison of the parameters of new oil with and without effective microorganisms.

| | Dynamic Viscosity in 2 °C | Flash Point | Acid Number | Base Number | Water Content |
|---|---|---|---|---|---|
| | [m·Pa·s] | [°C] | [mgKOH/g] | [mgKOH/g] | [%] |
| Fresh oil | 596 | 212.4 | 2.560 | 3.907 | 0.096 |
| Fresh oil and EM 2.5 mL | 720 | 200.4 | 3.021 | 3.606 | 0.210 |
| Fresh oil and EM 5 mL | 713 | 199.6 | 2.497 | 3.444 | above 1 |
| Fresh oil with EM 3 pcs | 667 | 212.5 | 3.113 | 3.962 | 0.072 |
| Fresh oil with EM 6 pcs | 696 | 213.5 | 2.366 | 3.894 | 0.097 |

In regards to the flash point shown in Figure 10, better results were also obtained for used oil with effective microorganisms than for those with additional silver compounds. The higher flash point of approximately 202 °C was obtained for effective microorganisms in the liquid form. The remaining additives have a flash point temperature below that of used oil, i.e., approximately 199 °C.

**Table 3.** Comparison of the parameters of fresh oil without additives with fresh oil with the addition of colloidal nanosilver and silver solution.

| | Dynamic Viscosity in 2 °C | Flash Point | Acid Number | Base Number | Water Content |
|---|---|---|---|---|---|
| | [m·Pa·s] | [°C] | [mgKOH/g] | [mgKOH/g] | [%] |
| **Fresh oil** | 596 | 212.4 | 2560 | 3.907 | 0.096 |
| **Fresh oil and colloidal nanosilver 2.5 mL** | 686 | 200.4 | 2.668 | 3.604 | above 1 |
| **Fresh oil and colloidal nanosilver 5 mL** | 742 | 198.4 | 2.659 | 3.751 | above 1 |
| **Fresh oil and silver solution 2.5 mL** | 698 | 206.9 | 2.682 | 3.479 | above 1 |
| **Fresh oil and silver solution 5 mL** | 736 | 201.2 | 2.660 | 3.731 | above 1 |

**Table 4.** Comparison of the parameters of used oil without additives with waste oil with effective microorganisms.

| | Dynamic Viscosity in 2 °C | Flash Point | Acid Number | Base Number | Water Content |
|---|---|---|---|---|---|
| | [m·Pa·s] | [°C] | [mgKOH/g] | [mgKOH/g] | [%] |
| **Used oil** | 719 | 198.8 | 4.027 | 1.358 | 0.167 |
| **Used oil and EM 2.5 mL** | 580 | 202.4 | 4.373 | 0.937 | 0.606 |
| **Used oil and EM 5 mL** | 758 | 202.1 | 5.411 | 0.844 | above 1 |
| **Used oil with EM 3 pcs** | 732 | 197.5 | 6.027 | 1.250 | 0.017 |
| **Used oil with EM 6 pcs** | 652 | 198.1 | 5.499 | 0.749 | 0.054 |

**Table 5.** Comparison of the parameters of waste oil without additives and waste oil with the colloidal nanosilver and silver solution.

| | Dynamic Viscosity in 2 °C | Flash Point | Acid Number | Base Number | Water Content |
|---|---|---|---|---|---|
| | [m·Pa·s] | [°C] | [mgKOH/g] | [mgKOH/g] | [%] |
| **Used oil** | 719 | 198.8 | 4.027 | 1.358 | 0.167 |
| **Used oil and colloidal nanosilver 2.5 mL** | 712 | 186.2 | 2.668 | 1.350 | above 1 |
| **Used oil and colloidal nanosilver 5 mL** | 758 | 194.0 | 2.659 | 1.402 | above 1 |
| **Used oil and silver solution 2.5 mL** | 796 | 191.6 | 2.682 | 1.037 | above 1 |
| **Used oil and silver solution 5 mL** | 806 | 197.6 | 2.660 | 1.199 | above 1 |

In regards to the fresh oil, statistical analysis was performed with a 95% significance level for used oil. The obtained results for water content and flash point for used oil and used oil with additives for this significance level also showed no significant errors and fell within the lower and upper confidence interval.

Similarly, from the graph of viscosity versus temperature (Figure 11), it can be seen that better results are obtained for effective microorganisms, also in liquid, because at the 2.5 mL for 2 °C, the viscosity is 588 mPa·s. This means that this value was lowered to a level similar to that of new oil, for which the viscosity is 596 mPa·s at the same temperature. For example, for colloidal nanosilver in a quantity of 2.5 mL, a viscosity of 1050 at 3.5 °C

was obtained, while the viscosity of the oil used at this temperature was 645 mPa·s. This means that the best additive to improve the properties of used oil is microorganisms in liquid form.

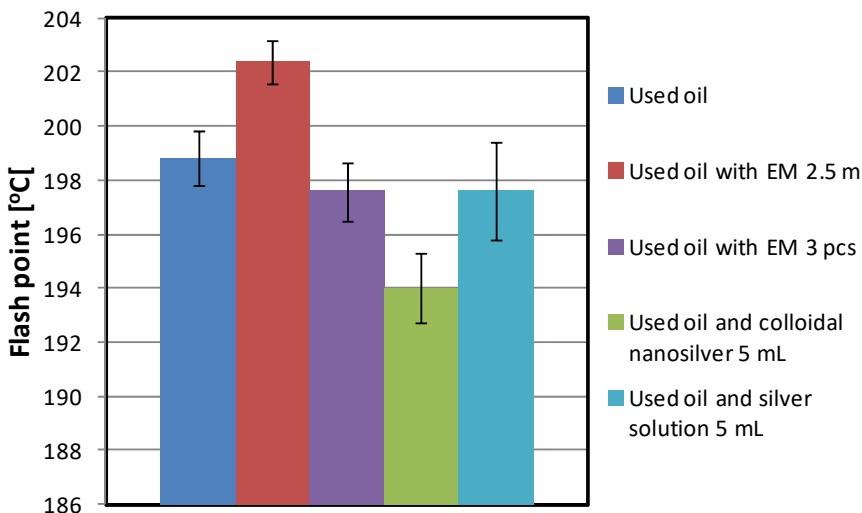

**Figure 10.** Oil flash point of waste oil and waste oil with effective microorganisms (EM), colloidal nanosilver (CN), and silver solution (SS).

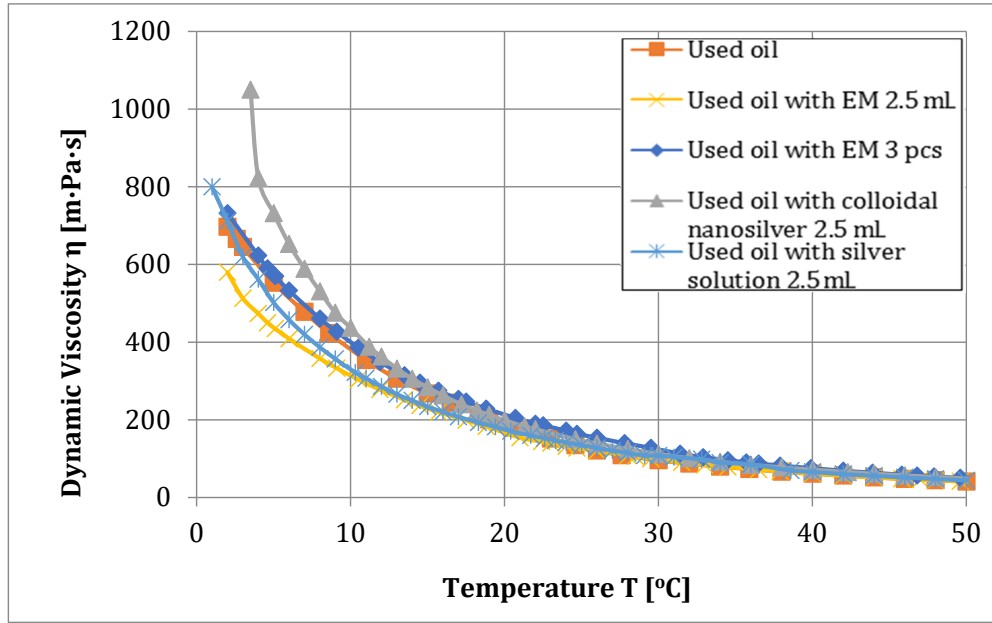

**Figure 11.** Comparison of viscosity of samples of used oil with and without the addition of effective microorganisms (EM), colloidal nanosilver (CN), and silver solution (SS).

## 5. Conclusions

In this paper, the influence of various environmentally friendly additives on the water content of lubricating oil was presented.

For fresh oil, the water content was 0.096%. Liquid effective microorganisms in a quantity of 2.5 mL, taken from a 1 L package, resulted in a more than twofold increase in the water content in the oil, while 5 mL of the solution of effective microorganisms resulted in the over-titration of the device, i.e., the water content exceeded 1%.

Silver solution and colloidal nanosilver in amounts of 2.5 mL and 5 mL were also added to the fresh oil. In this case, it was not possible to read the result of the water content in the oil for both the first and the second type of silver and for each amount, because after

prolonged operation the titrator showed over-titration, i.e., the water content in the oil exceeded the measuring range of the device. The reason for this is also too much water in the additive itself, which is a silver solution.

The situation is different for the addition of EM ceramic tubes, wherein which the microorganisms are bound in the clay and there is no water in the composition of the ceramic tube. Ceramic tubes in an amount of three pieces reduced the water content in the new oil, which may have a good effect on the parameters of the oil, as it will cause the subsequent deterioration of the properties of the oil operating in the internal combustion engine. As can be seen from the charts obtained, an increase in the number of ceramic tubes did not result in a further reduction of the water content, so in further research it is worth examining what parameters would be obtained with fewer than three tubes.

The analysis of the water content graphs for used oil with and without the addition of effective microorganisms, colloidal nanosilver, and silver solution show that the effect of additives presents similar results to those of new oil.

In the used oil, the water content was about 0.17%, while for microorganisms in liquid form added to the oil in an amount of 2.5 mL, the water content was 0.6%; for 5 mL the result was not obtained due to excessive titration. The same was the case with fresh oil. This means that the water amount was above the measuring range. The water content for three ceramic tubes was about 0.02%, for six tubes it was 0.05%, and for fresh oil the water content was 0.09%. The addition of silver, regardless of its type and amount, resulted in an amount of water greater than 1%.

To conclude, in practice the most effective and recommended type of engine oil treatment is the use of effective microorganisms in the form of ceramic tubes. In future research, the influence on the parameters of the oil will be investigated with a smaller number of ceramic tubes, which will then be used within the internal combustion engine, in conditions similar to real ones. This will make it possible to verify the current research and more precisely determine which additives may be better used to ameliorate the parameters of the engine oil.

**Funding:** This research received no external funding.

**Institutional Review Board Statement:** Not applicable.

**Informed Consent Statement:** Not applicable.

**Data Availability Statement:** The data used to support the finding of this study are available from the corresponding author upon request.

**Conflicts of Interest:** The author declares no conflict of interest.

### Nomenclature

| | |
|---|---|
| ACE | Automatic Error Compensation |
| ACF | Activated Carbon Fiber |
| DNA | Deoxyribonucleic Acid |
| EM | Effective Microorganisms |
| HTHS | High Temperature High Shear Rate |
| SAE | Society of Automotive Engineers |
| SS | Silver solution |
| CN | Colloidal nanosilver |

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
