# Peer review of "Influence of Effective Microorganisms, Colloidal Nanosilver and Silver Compounds on Water Content in New and Used Engine Oil: A Preliminary Study"

_applsci, doi:10.3390/app122010234_

Round 1

Reviewer 1 Report (New Reviewer)

·        Fig. 1 and 2: I recommend that instead of a photo of the commercial packaging, the manufacturer, origin and technical data should be presented.

·        Fig. 5 should be Fig. 3; Fig. 6 should be Fig. 4 and so on…

·        „Results and analysis of research“ should be "Results and Discussion"

·        The description of the 5th pillar of Fig. 9 is missing, probably  “silver solution”.

·        Page 7: “This viscosity is 720 mPa∙s at approx. 2 oC, which is similar to used oil, for which the viscosity at this temperature is 719 mPa∙s. Other additives, including silver compounds, also increase the viscosity of the oil, but not as much as effective microorganisms in liquid form. What kind of used oil are we talking about? In Fig. 10 there is no curve corresponding to used oil.

·       In my opinion, Fig. 10 – 13 present some results in two different form (Tables and Figures): For example Fig. 12 presents results from Table 3 (flash point)

·        Some results from Tables 1- 4 are similar to those presented in [55-57]:

Krakowski, R. Research on the effect of the effective microorganisms, silver solution and colloidal nanosilver addition on the engine oil acid number (TAN), Combustion Engine 2021, 186 (3), 59-63. Available online: http://www.combustion-engines.eu/Research-on-the-effect-ofthe-effective-microorganisms-silver-solution-and-colloidal,140730,0,2.html (accessed on 12 July 2022). 56. Krakowski, R. Research on the effect of the effective microorganisms, silver solution and colloidal nanosilver addition on the engine oil base number (TBN), Combustion Engine 2021, 187 (4), 8-11. Available online: http://www.combustion-engines.eu/Research-on-the-effect-ofthe-effective-microorganisms-silver-solution-and-colloidal,140112,0,2.html (accessed on 12 July 2022).Krakowski, R. Research into the effects of the effective microorganisms addition on the engine oil viscosity Journal of KONES 2019, 26 (3), 105-112. Available online: https://kones.eu/ep/2019/vol26/no3/105-112_JOK_2019_NO.3_VOL.26_ISSN_1231-4005- KRAKOWSKI.pdf

The conclusion is too long. It would be good if the results were more general.

Author Response

Review 1

Fig. 1 and 2: I recommend that instead of a photo of the commercial packaging, the manufacturer, origin and technical data should be presented.

Basic information about these additives is presented in the article. As for Fig. 1 and Fig. 2, they were included in the article at the suggestion of one of the previous Reviewers. The manufacturer has been addend.

Fig. 5 should be Fig. 3; Fig. 6 should be Fig. 4 and so on…

It has been corrected.

  „Results and analysis of research“ should be "Results and Discussion"

It has been corrected.

The description of the 5th pillar of Fig. 9 is missing, probably  “silver solution”.

It has been corrected.

Page 7: “This viscosity is 720 mPa∙s at approx. 2 oC, which is similar to used oil, for which the viscosity at this temperature is 719 mPa∙s. Other additives, including silver compounds, also increase the viscosity of the oil, but not as much as effective microorganisms in liquid form.”  What kind of used oil are we talking about? In Fig. 10 there is no curve corresponding to used oil.

It has been corrected.

 In my opinion, Fig. 10 – 13 present some results in two different form (Tables and Figures): For example Fig. 12 presents results from Table 3 (flash point)

True, these are the same results, but the data in the tables are presented as a compilation and comparison of previously obtained results to confirm the obtained results. The aim of the drawing (flash point) is to enrich the article, while the flash point value in the table together with other results summarizes the results obtained so far.

Some results from Tables 1- 4 are similar to those presented in [55-57]:

It is true that some results from Tables 1- 4 are similar to those presented in [55-57], because the results in the tables are taken from those articles to compare and confirm the results. The tables have the source where they come from, that is [55-57]. This was required by the previous Reviewer.

The conclusion is too long. It would be good if the results were more general.

Where possible, the conclusion have been shortened. In the previous review, the Reviewers wrote that the conclusions were too general, that they did not contain quantitative results, and I was to improve it, hence the final shape of these conclusions.

Reviewer 2 Report (New Reviewer)

Dear authors

the contribution deals with the issue of the effect of the effective microorganisms and silver compounds addition on the water content of new and used oil. I have the following comments and suggestions for improvement on the article:

- The methodology of the experiment could be clearer, for example written in bullet points

- a more detailed description of measuring devices (accuracy, range, etc.) is missing

- was the number of repetitions determined by statistics?

- what type of base oil is it (PAO or other) is it good to mix with your improvers?

- can effective microorganisms and silver compounds influence on additives in the oil?

- there is no discussion (it would be appropriate to compare your results with similar research)

- why weren't a smaller number of tubes used?

Thank you and have a nice day.

Author Response

Rewiev 2

The methodology of the experiment could be clearer, for example written in bullet points

Suggestion was taken into account.

- a more detailed description of measuring devices (accuracy, range, etc.) is missing

It has been corrected.

- was the number of repetitions determined by statistics?

There is information in the paper that each obtained result was repeated 3 times and the result presented in the graph in the article is the arithmetic mean of these measurements.

 - what type of base oil is it (PAO or other) is it good to mix with your improvers?

The article contains information about what oil it is:

5W30 synthetic oil was used for the tests, which slightly changes its viscosity during temperature changes. As a result, it perfectly lubricates engine parts, while protecting it against deposits of carbon deposits, sludge and other harmful impurities that can accelerate its wear.

The described oil retains its optimal viscosity from -30 oC and its fluidity at -35 oC. 5W30 oil has high pumpability, thanks to which it is highly effective in cleaning the most important engine components, while contributing to their efficient cooling [37].

5W30 automotive oils according to the SAE J300 standard of 2015, must meet the following criteria [38]:

  • maximum dynamic viscosity 6600 cP at -30 oC,
  • maximum pumpability temperature of 60,000 at the temperature of. -35 oC,
  • kinematic viscosity at temperature 100 oC min. 3,8 mm2/s to 9,3 - 12,5 mm2/s,
  • viscosity HTHS in 150 oC 2,9 cP.

Earlier tests (the results of which are presented in Tables 2 ÷ 5 did not show a negative effect on the parameters of the oil, despite the content of other enriching additives.

- can effective microorganisms and silver compounds influence on additives in the oil?

This is the subject of my research. The results obtained so far show that the addition of effective microorganisms in the form of ceramic tubes has the most beneficial effect on the parameters of the oil.

- there is no discussion (it would be appropriate to compare your results with similar research)

So far, effective microorganisms have been used in wastewater treatment, in horticulture, and silver compounds in combating microbes from food or treating people, so at the moment it is difficult to discuss with regard to petroleum products.

- why weren't a smaller number of tubes used?

The number of tubes selected was chosen on the basis of the amounts used in other industries (e.g. wastewater treatment, agricultural crops). Initial tests are currently being carried out and based on the obtained results, it has been found that the tests should be repeated with a smaller number of ceramic tubes (EM).

Reviewer 3 Report (New Reviewer)

The work entitled “Effect of effective microorganisms, colloidal nanosilver and silver compounds addition on the water content in new and used engine oil: a preliminary study, the only author is Rafał Krakowski, is devoted to the problem of water accumulation in engine oils. Increased water content in engine oils can provide a strong friction, corrosion and result in accelerated engine wear. The author suggested to use additives on the base of microorganisms or silver to reduce water content in engine oils. Although, the studied problem is really important, the efficacy of suggested approaches does not seem significant, and experimental methods and interpretation of results do not look valid. In its current form, this paper is not suitable for publication in the journal Applied Sciences.

Major revisions

1) The title is very long and excessive. The word combination “Effect of effective” is not informative. It is recommended to shorten the title, for example like “Effect of microorganisms, colloidal nanosilver and silver compounds on water content in new and used engine oil”.

2) It is strongly recommended to show this paper to any microbiologist. There are some inaccuracies:

- the term “effective microorganisms” is not used in microbiology; it is more correctly to use one of these terms “a microbial association”, “a microbial consortium”, or “a microbial preparation”;

- all Latin names including names of genera and species epithets must be italicized; in case species epithets are not indicated, Saccharomyces spp. and Streptomyces spp. must be written;

- manufacturers of the microbial preparations used (Fig.1) must be indicated in Materials and Methods; photos are not sharp, and labels on bottles are not readable;

- What were concentrations of microorganisms in all experiments? It is important since quantity of water in liquid preparations is less in concentrated microbial suspensions and more in diluted suspensions, and risks for corrosion and microbial oxidation of the engine oil with its following deterioration depends on number of metabolically active microorganisms. It is recommended to give these cell concentrations (usually as CFU/ml) in Fig. 6, 9, and 11 instead of volumes 2.5 or 5.0 ml;

- Were some sterile controls (sterilized engine oil in sterilized glassware) used? Maybe, the reasons for revealed positive effects in reducing water content in the engine oil are unidentified microorganisms from air, oil itself, or glassware;

- How ceramic tubes with microorganisms are obtained? Where are microorganisms located? On the surfaces of ceramic tubes, or somewhere in pores or channels, with which (probably) the tubes were perforated? It is important to understand the level of availability of nutrients, gases and water solubilized minerals to microbial cells.

3) In Materials and Methods, additional significant inaccuracies are found, and many details are lost (all these details are important; otherwise, other researchers, most probably, cannot repeat described experiments and obtain the same effects from oil additives):

- the manufacturer for the 5W30 synthetic oil is not indicated;

- it is confusing: the author writes that silver is used in two forms, as silver compounds and as colloidal silver nanoparticles. Silver compounds are not indicated. Are they some salts, what salts, in what concentration are they applied? Concentrations for colloidal silver nanoparticles are not indicated too. Also, it is written that nanosilver visible as particles under an electronic microscope is silver ions but ions mean salts dissolved in water, and dissolved substances hardly form particles. Similar to other reagents, manufacturers for colloidal nanosilver and unnamed silver compounds are not indicated in the paper. Procedure for production of colloidal nanosilver is not described;

- it is unclear how the used oil is obtained: this procedure is not described. Was it oil after one, two or more operation cycle(s)? In case it was not artificially modeled used oil, how used oil was standardized in various experimental trials?

- Did the author take into account that liquid forms of microbial and silver additives added to total volume of water? Approximate water content in the engine oils with additives must be as high as 2.5 or 5.0%. But it is less than 1%, apart from Fresh oil + EM 5 ml (Fig. 6 and 9). How it could be explained?

4) In Fig. 6, 9, 11, and 12, errors (standard deviations, or confidential intervals) and/or p-values must be indicated. Are revealed differences statistically significant?

5) What are possible mechanisms explaining why microbial and silver additives reduce water content in engine oil? Concerning silver, its antimicrobial (not water eliminating) properties are mentioned only.

6) Why water content was measured in used oil? If oil is used, there are no reasons to increase its quality and eliminate water.

7) Why water content in used oil with additives is high (more than 1%) in comparison with fresh oil, again apart from Fresh oil + EM 5 ml? Could it be related with increased water absorption by used oil?

Minor revisions

In the text, references to Fig. 7 and 8 are made before references to Fig. 1, 2 … Maybe, it is reasonable to change figure enumeration?

In description for Fig. 11 in the text, effects in used oil are discussed. However, only data for fresh oil are presented in Fig. 11.

Summary

Reject

Author Response

Rewiev 3

Major revisions

1) The title is very long and excessive. The word combination “Effect of effective” is not informative. It is recommended to shorten the title, for example like “Effect of microorganisms, colloidal nanosilver and silver compounds on water content in new and used engine oil””.

I would like to propose a different change, because the word effective is the key in this case, it is the full name of these microorganisms and the exclusion of this word does not fully reflect what microorganisms are meant:

“Influence of effective microorganisms (EM), colloidal nanosilver and silver compounds on water content in new and used engine oil: a preliminary study”.

2) It is strongly recommended to show this paper to any microbiologist. There are some inaccuracies:

- the term “effective microorganisms” is not used in microbiology; it is more correctly to use one of these terms “a microbial association”, “a microbial consortium”, or “a microbial preparation”;

There is a proper name proposed by prof. Teruo Higa. Hence the term in the article.

https://www.emrojapan.com/

- all Latin names including names of genera and species epithets must be italicized; in case species epithets are not indicated, Saccharomyces spp. and Streptomyces spp. must be written;

It has been corrected.

- manufacturers of the microbial preparations used (Fig.1) must be indicated in Materials and Methods; photos are not sharp, and labels on bottles are not readable;

It has been added to the article.

What were concentrations of microorganisms in all experiments? It is important since quantity of water in liquid preparations is less in concentrated microbial suspensions and more in diluted suspensions, and risks for corrosion and microbial oxidation of the engine oil with its following deterioration depends on number of metabolically active microorganisms. It is recommended to give these cell concentrations (usually as CFU/ml) in Fig. 6, 9, and 11 instead of volumes 2.5 or 5.0 ml;

It has been added to the article.

 Were some sterile controls (sterilized engine oil in sterilized glassware) used? Maybe, the reasons for revealed positive effects in reducing water content in the engine oil are unidentified microorganisms from air, oil itself, or glassware;

The sterile place was the titration cell - an electrolyser with a double sealing system:

cut + intermediate gasket - for maximum

system tightness, i.e. during the measurement itself. Before the measurement, the oil was stored in closed bottles in a dark place.

- How ceramic tubes with microorganisms are obtained? Where are microorganisms located? On the surfaces of ceramic tubes, or somewhere in pores or channels, with which (probably) the tubes were perforated? It is important to understand the level of availability of nutrients, gases and water solubilized minerals to microbial cells

Effective microorganisms in the form of ceramic tubes are a special type of clay inoculated with beneficial microflora (fermented clay), which mature in natural conditions for several months, then it is fired under high temperatures in anaerobic conditions, compositions of beneficial microorganisms, sugar cane molasses, restructured water. Clays inoculated with microorganisms mature for many weeks in appropriate humidity and temperature, and then, depending on their intended use, they are fired at high temperatures (gray) of 1200 - 1300º. When fired, they become a viscous liquid, and when cooled to ambient temperature, they bind strongly and are extremely hard. Ceramics fired thanks to this method are less porous, denser and less brittle. The raw materials used in traditional ceramics are minerals such as clay, silica stone, and feldspar. Ceramics are inorganic and contain no metals. It was shaped and then hardened by firing at high temperatures. Ceramics do not react chemically, do not burn and are harder than steel. The shapes of the fired elements or dishes are adjusted to their future use. As in the liquid form, the content of effective microorganisms is 3% in each ceramic tube.

3) In Materials and Methods, additional significant inaccuracies are found, and many details are lost (all these details are important; otherwise, other researchers, most probably, cannot repeat described experiments and obtain the same effects from oil additives):

- the manufacturer for the 5W30 synthetic oil is not indicated;

It has been added to the article.

 - it is confusing: the author writes that silver is used in two forms, as silver compounds and as colloidal silver nanoparticles. Silver compounds are not indicated. Are they some salts, what salts, in what concentration are they applied? Concentrations for colloidal silver nanoparticles are not indicated too. Also, it is written that nanosilver visible as particles under an electronic microscope is silver ions but ions mean salts dissolved in water, and dissolved substances hardly form particles. Similar to other reagents, manufacturers for colloidal nanosilver and unnamed silver compounds are not indicated in the paper. Procedure for production of colloidal nanosilver is not described;

Silver solution

Ion silver is as transparent as water. If the manufacturer additionally indicates that silver must be stored in glass bottles, it means that it is not real colloidal silver. Real silver colloids are lightfast and therefore do not need to be stored in glass.

Ionic silver contains 90% silver ions and only about 10% silver particles. As 90% of the particles are silver ions, a more appropriate term would be "silver solution". There is no such thing as ionic silver particles in science. Molecules are not ions and silver ions are not particles. These terms are not unambiguous, are not synonymous and cannot be used interchangeably. After entering the body, silver ions immediately combine with it, creating an insoluble salt - silver chloride. Such a property is not exhibited by electrically neutral silver particles. Silver chloride does not have any properties that are attributed to colloidal silver.

Colloidal nanosilver

Due to the high concentration of silver nanoparticles, colloidal silver does not resemble water. Its characteristic feature is a clear color, which can range from yellow to amber. Colloidal silver has a darker color because the silver particles dispersed in the water block the light passing through them with a specific wavelength, which is approx. 400 nm. Additionally, it can be stored in plastic bottles because silver nanoparticles are chemically insensitive and completely resistant to the damaging effects of UV radiation.

Colloidal silver with silver nanoparticles is the least common type of silver commercially due to its complicated process and high production costs. However, it is the only effective silver with scientifically proven properties. Real colloidal silver consists of 80% silver particles and the remaining 20% ​​silver ions. The decisive factors for the quality of real colloidal silver are the content of silver particles and their very active total surface. In colloidal silver, nanometer-sized particles form a colloid, so it is not necessary to use a protein substrate to suspend them in water. Due to the electrokinetic potential, nanoparticles repel each other, keeping the colloid stable.

The silver concentration value and manufacturers were added to the article.

it is unclear how the used oil is obtained: this procedure is not described. Was it oil after one, two or more operation cycle(s)? In case it was not artificially modeled used oil, how used oil was standardized in various experimental trials?

It has been added to the article.

 Did the author take into account that liquid forms of microbial and silver additives added to total volume of water? Approximate water content in the engine oils with additives must be as high as 2.5 or 5.0%. But it is less than 1%, apart from Fresh oil + EM 5 ml (Fig. 6 and 9). How it could be explained?

The permissible water content in new oil is less than 1000 ppm (0.1%). And that's all that came out in my research.

4) In Fig. 6, 9, 11, and 12, errors (standard deviations, or confidential intervals) and/or p-values must be indicated. Are revealed differences statistically significant?

It has been added.

5) What are possible mechanisms explaining why microbial and silver additives reduce water content in engine oil? Concerning silver, its antimicrobial (not water eliminating) properties are mentioned only.

The article does not say that all additives reduce the water content, on the contrary, due to over-titration (water content too high) no graphs were made for the addition of silver. The addition of effective in liquid form also caused an increase in the water content in the oil, only microorganisms and in a smaller amount (3 pieces) reduced the amount in the oil. A possible mechanism is described in the note below.

6) Why water content was measured in used oil? If oil is used, there are no reasons to increase its quality and eliminate water.

This was done for comparative purposes to see how much water is in the used oil compared to the new oil. It is also checked how these additives work in the used oil, whether it makes sense to add them during the service life to extend the time until the next change.       

7) Why water content in used oil with additives is high (more than 1%) in comparison with fresh oil, again apart from Fresh oil + EM 5 ml? Could it be related with increased water absorption by used oil?

As in the new oil, the greatest amount of water occurs when effective microorganisms are added in liquid form. It can be seen from the diagram that the more additive (5 ml), the water content reaches very high values (above 1%). When microorganisms are added in the form of ceramic tubes, the amount of water in the used oil is even lower than in the new oil. This is due to the very action of effective microorganisms and the structure of the ceramic tubes. It is also worth checking the water content with smaller amounts of ceramic tubes, because due to the fact that there is also bound water, too many tubes do not have a positive effect on reducing the water content in the oil.

In the text, references to Fig. 7 and 8 are made before references to Fig. 1, 2 … Maybe, it is reasonable to change figure enumeration?

It has been corrected.

In description for Fig. 11 in the text, effects in used oil are discussed. However, only data for fresh oil are presented in Fig. 11.

It has been corrected.

Reviewer 4 Report (Previous Reviewer 3)

The author addressed all the comments and revised his manuscript in a very effective way. Now, the manuscript presents an interesting study of an important application and deserves to be published.

Author Response

Dear Reviewer, thank you very much for the positive assessment

Round 2

Reviewer 1 Report (New Reviewer)

The author has taken into account my remarks.

Author Response

Dear Reviewer, thank you very much for the positive assessment

Reviewer 3 Report (New Reviewer)

The work entitled “Influence of effective microorganisms (EM), colloidal nanosilver and silver compounds on water content in new and used engine oil: a preliminary study, the only author is Rafał Krakowski, is devoted to the problem of water accumulation in engine oils. The author has corrected the paper after the reviewers’ comments. However, significant details still remain unexplained.

Specific comments

1) Abbreviations are rarely used in titles. It is not necessary to give abbreviation EM in the Title.

2) p. 3, paragraph “Effective microorganisms are specially selected…” – “Saccharomyces” should be corrected as “Saccharomyces spp.”

3)  p. 3, paragraph “Effective microorganisms are specially selected…” – why Streptomyces is written twice? Does it mean “Streptomyces spp.”?

4) p. 3, paragraph “The principle of operation of EM…” – according to the description in this paragraph, inoculated ceramics was heated at high temperature as 1,200–1,300°C after inoculation with microorganisms. Microorganisms are died at such high temperature. Does it mean that effect of EM is related with microbial cell components, not with living cells? Moreover, at this temperature all organics is burnt. Does it mean that effect from EM is more related even with changes in mineral composition of ceramics? It means also that liquid and ceramic adhered EM have various mechanisms of action: in liquid EM, living cells act, and in ceramics, ceramic tubes as non-living objects act.

5) p. 3 – the concentration of EM as 3% is indicated for all experiments. However, it is not clear, how many cells were introduced?

6) p. 4, paragraph “Experimental data from studies…” – Staphylococcus aureus must be italicized.

7) Figure 4 and Table 2, Figure 7 and Tables 3, Figure 9 and Table 8, and Figure 10 and Table 9 – these pairs of figure + table present same information. It is recommended to include in the paper only figure or only table from each pair. In Tables 2,3,8, and 9, it is not necessary to show results for each sample, average ± standard deviation is enough. Moreover, in Figures, it is strongly recommended to show statistics.

8) In Conclusion, it is recommended to summarize what type of engine oil treatment is most effective and recommended for practice.

9) Possible applications of the obtained results remain unclear. Did author try to reduce water content or eliminate unwanted microflora in engine oils?

Summary

Major revision

Author Response

Rewiev  3

Specific comments

1) Abbreviations are rarely used in titles. It is not necessary to give abbreviation EM in the Title.

It has been corrected.

2) p. 3, paragraph “Effective microorganisms are specially selected…” – “Saccharomyces” should be corrected as “Saccharomyces spp.”

It has been corrected.

3)  p. 3, paragraph “Effective microorganisms are specially selected…” – why Streptomyces is written twice? Does it mean “Streptomyces spp.”?

It has been corrected.

4) p. 3, paragraph “The principle of operation of EM…” – according to the description in this paragraph, inoculated ceramics was heated at high temperature as 1,200–1,300°C after inoculation with microorganisms. Microorganisms are died at such high temperature. Does it mean that effect of EM is related with microbial cell components, not with living cells? Moreover, at this temperature all organics is burnt. Does it mean that effect from EM is more related even with changes in mineral composition of ceramics? It means also that liquid and ceramic adhered EM have various mechanisms of action: in liquid EM, living cells act, and in ceramics, ceramic tubes as non-living objects act.

EM ceramics is a separate chapter of the Effective Microorganism technology. It was very interesting for Professor Higa to learn that in 1967 scientists from NASA found an ordinary lactic acid bacterium Strepoccocus mitis, quite healthy and ready to multiply, in a device that returned from a two-year trip to the Moon. The bacteria survived in extreme temperatures, in an anaerobic environment, in an observation camera - right inside the lens housing. Researching the resistance of anaerobic bacteria to high temperatures, prof. Higa found that the ceramic vessel retained the EM properties for several months. According to the theory of prof. Higi's effective microorganisms are trapped in pottery as living things. This insight led the professor to develop a method of burning EM-fermented clay in an anaerobic process. In the clay, microorganisms are trapped, which multiply and drag similar ones to each other. This permanent "residence" can solve the problem of migration of microorganisms contained in EM in liquid form. More information in the link below:

https://www.emnatuurlijkactief.nl/wp-content/uploads/2014/04/Ceramics-brochure-for-the-website.pdf

5) p. 3 – the concentration of EM as 3% is indicated for all experiments. However, it is not clear, how many cells were introduced?

I do not know how many cells have been introduced, because the manufacturer does not provide such information.

6) p. 4, paragraph “Experimental data from studies…” – Staphylococcus aureus must be italicized.

It has been corrected.

7) Figure 4 and Table 2, Figure 7 and Tables 3, Figure 9 and Table 8, and Figure 10 and Table 9 – these pairs of figure + table present same information. It is recommended to include in the paper only figure or only table from each pair. In Tables 2,3,8, and 9, it is not necessary to show results for each sample, average ± standard deviation is enough. Moreover, in Figures, it is strongly recommended to show statistics.

8) In Conclusion, it is recommended to summarize what type of engine oil treatment is most effective and recommended for practice.

It has been added.

9) Possible applications of the obtained results remain unclear. Did author try to reduce water content or eliminate unwanted microflora in engine oils?

I did not try to reduce water content or eliminate unwanted microflora in engine oils. In further research, I will reduce the amount of the addition of effective microorganisms.

This manuscript is a resubmission of an earlier submission. The following is a list of the peer review reports and author responses from that submission.

Round 1

Reviewer 1 Report

Author should be more explicit about how their innovation is different from others.

The abstract is too general: it no contains quantitative results. The conclusion also should be rewritten.

-The “Introduction”, the author needs to collect the scholars from all over the world to study the antibacterial effect, lubricating oil water content and lubricating oil detection, the influence of lubricating oil additives on oil products, that must present research from various counties of the world to represent the worldwide background of the submitted manuscript.

-The whole article obviously still expresses the influence of water on lubricating oil, but the author’s understanding of lubricating oil is slightly insufficient.

-The author’s lack of knowledge about the stability and stability of lubricants.

-Generally, the lubricating oils have standard methods for testing oil products, and the author’s research is more like a basic analysis of biodiesel.

-The author's setting and definition of the lubricating oil aging experiment is very arbitrary, resulting in a lack of research value for the overall study.

-Author uses colloidal nanosilver and silver solution to add antibacterial to lubricating oil, which is basically a wrong model, because the maximum Impact factor affecting lubricating oil are water and temperature. Adding colloidal nanosilver and aqueous solution will directly affect lubrication.

-Overall, read carefully the whole article several times, the author's understanding of the antibacterial effect, lubricating oil water content and lubricating oil detection, the influence of lubricating oil additives on oil products seems very strange and ignorant. The discussion on their research method is very lacking and not in-depth.

Author Response

 Review 1

-Author should be more explicit about how their innovation is different from others.

          The problem of the risk of microbial contamination of diesel fuel and the use of biocides is known and well described in the literature. The problem of microbiological contamination of engine oils discussed in the paper is not widely known. Biocides are pesticides used, inter alia, to combat or limit the growth of microorganisms in petroleum products. Pestcides are harmful to the environment, so I would like to propose the use of other environmentally friendly products, instead of harmful biocides.

-The abstract is too general: it no contains quantitative results. The conclusion also should be rewritten.

The abstract has been improved and is more detailed, it also includes the purpose of the research and the results obtained in the research. The conclusions were also rebuilt, more elaborate and contain quantitative results from the obtained research.

-The “Introduction”, the author needs to collect the scholars from all over the world to study the antibacterial effect, lubricating oil water content and lubricating oil detection, the influence of lubricating oil additives on oil products, that must present research from various counties of the world to represent the worldwide background of the submitted manuscript.

The introduction describes the problem of microbial contamination in petroleum products. The current state of research on this issue is presented. Currently used methods of preventing microbiological contamination of petroleum products are presented. A new alternative method of preventing microbial contamination or reducing the effects of this phenomenon using environmentally friendly measures has also been proposed. This is described in more detail in the author's earlier articles:

Problems of contamination occurring in petroleum products and their removal by means of environmentally friendly means, Autobusy. Technika, Eksploatacja, Systemy Transportowe
 2017, r. 18 (No 6 (208)), pp. 275-280, In Polish

Available online: http://yadda.icm.edu.pl/baztech/element/bwmeta1.element.baztech-6c529361-a80b-4eae-88f9-e4cba5adad15?q=bwmeta1.element.baztech-e687c8e6-d61e-4483-abfd-7305024c9880;49&qt=CHILDREN-STATELESS

Microbiological contamination of diesel and lubricating oils and the possibility of its reduction, Autobusy. Technika, Eksploatacja, Systemy Transportowe 2016, 17 (12), pp. 292 – 296, In Polish.

Available online: http://www.autobusy-test.com.pl/images/stories/Do_pobrania/2016/nr%2012/Bezpieczenstwo/49_190_A_BiE_KRAKOWSKI.pdf

-The whole article obviously still expresses the influence of water on lubricating oil, but the author’s understanding of lubricating oil is slightly insufficient.

The article presents the obtained research on the effect of the addition of effective microorganisms and nanosilver on the water content in the oil. This is only one of the examined issues, everything cannot be presented in such a short material, hence the impression that the author does not understand lubricating oil. Some of the results have already been published and the author conducts further research on this issue.

-The author’s lack of knowledge about the stability and stability of lubricants.

The stability of lubricants has not been investigated, but the author will consider this in further research.

-Generally, the lubricating oils have standard methods for testing oil products, and the author’s research is more like a basic analysis of biodiesel.

The tests are carried out in accordance with the requirements of the ASTM D6304 standard. There is a method for determining water in products petroleum, lubricating oils and adhesives by titration coulometric by Karl Fisher. The oil used for the tests was not biodiesel oil. More about the type and properties of the tested oil can be found in the revised article.

-The author's setting and definition of the lubricating oil aging experiment is very arbitrary, resulting in a lack of research value for the overall study.

The article does not present the aging process of the oil. The article is about replacing environmentally harmful additives with environmentally friendly ones in order to prevent the development of microbial contamination in new and used oil.

-Author uses colloidal nanosilver and silver solution to add antibacterial to lubricating oil, which is basically a wrong model, because the maximum Impact factor affecting lubricating oil are water and temperature. Adding colloidal nanosilver and aqueous solution will directly affect lubrication.

Colloidal silver was used in the research because it has primarily antibacterial, fungicidal and virucidal properties. This was done in order to compare which of the additives would be better for the properties of the engine oil. Currently, the oil with these additives has been used in the working engine and the next tests will be aimed at assessing the influence of these additives on the properties of the oil and the technical condition of the internal combustion piston engine.

-Overall, read carefully the whole article several times, the author's understanding of the antibacterial effect, lubricating oil water content and lubricating oil detection, the influence of lubricating oil additives on oil products seems very strange and ignorant. The discussion on their research method is very lacking and not in-depth.

The article has been thoroughly edited, rebuilt and supplemented with the missing information, taking into account the comments provided by the Reviewer

Reviewer 2 Report

The paper submitted for review shows the relationship between the introduction of microorganisms, colloidal nanosilver and silver into engine oil and the water content in the oil. Contamination of water can pose a number of problems and therefore this kind of research is of interest and practical importance. Unfortunately, the reviewed manuscript is not well written and needs to be significantly improved.

Abstract

Needs to be rewritten. The aim and scope of the study should be indicated, a summary of the results obtained and conclusions should be provided.

Introduction

This chapter does not provide an introduction to the topic of the research carried out and needs to be rewritten. I suggest to remove the subsections. Incorrectly cited references - should start with citation 1 etc., not alphabetically. Line 45 - desinfection is a chemical method of sterilisation. Not physical disinfection - this is physical, thermal sterilisation. Too much content on characteristics of lubricating oil, water contamination. Repetition should be avoided, e.g. about the use of oil components by microorganisms as a source of carbon and energy, about the forms of water in oils. Incorrect designation of degrees Celsius, not only in this chapter. In subsection 3.1, the entire text (several paragraphs) is information from 4 references given in line 130-131? Remove Fig. 1 and text from line 136-147, possibly use a synthetic description of this section.

Author did not refer in any way to the topic of his research. No information related to the use of oil additives. Nothing about effective microorganisms. This section should include a clear research hypothesis.

"The research stand and methodology"

This chapter should be changed to Materials and Methods and prepare again. Describe the research material - characteristics of the oils, microorganisms (ready-to-use preparation, consortium?), used additives and describe the experimental scheme. Figure 2 is unnecessary. Remove description from lines 167-175 (why the construction of the analyser?). Shorten description in subsection 4.1 - only describe the determination methodology. Figure 3 - delete. Test from line 192-202 delete.

No information about number of repetitions for analyses, elementary statistics. Nothing known about the EM preparation, the solution they were in (medium, composition), how were the microorganisms deposited on the ceramic tubes? What did the tubes look like etc.

Chapter 5 in the paper should be results, not conclusions

In this chapter Author presents the scheme of the study, e.g. the amount of introduced microorganisms - this is not this chapter. Fig. 4 should be marked EM 5 mL object, with a relevant description about the lack of analysis, can be in the signature. It was not the microorganisms that increased the water content but the solution they were in and which was introduced into the oil. Water does not mix readily with oil due to its polar nature and the non-polar nature of hydrocarbons. So why was an emulsion formed? Line 265 "697" - where did this data come from? Figure 9 - complete the axis description " The volume... in the oil". Line 288 "...smaller Mount". Tables 1- 4 contain a lot of interesting information, but Author did not describe them. In tables - same number of decimal places. Figure 7 on page 12 is Fig. 10, bracketed.

Lack of statistical analysis complicates conclusions.

Author should discuss the results!!!

Conclusions need to be rewritten.

References

Should be amended as required by the Editor.

Author Response

Review 2

Abstract

Needs to be rewritten. The aim and scope of the study should be indicated, a summary of the results obtained and conclusions should be provided.

Abstract has been rewritten. The purpose and scope of the study has been described. The conclusions were also rebuilt, more elaborate and contain quantitative results from the obtained research.

Introduction

This chapter does not provide an introduction to the topic of the research carried out and needs to be rewritten.

This chapter has been changed.

I suggest to remove the subsections.

The subsections has been removed.

Incorrectly cited references - should start with citation 1 etc., not alphabetically.

Cited references have been corrected

Line 45 - desinfection is a chemical method of sterilisation. Not physical disinfection - this is physical, thermal sterilisation. Too much content on characteristics of lubricating oil, water contamination. Repetition should be avoided, e.g. about the use of oil components by microorganisms as a source of carbon and energy, about the forms of water in oils. Incorrect designation of degrees Celsius, not only in this chapter. In subsection 3.1, the entire text (several paragraphs) is information from 4 references given in line 130-131? Remove Fig. 1 and text from line 136-147, possibly use a synthetic description of this section.

That all has been corrected.

Author did not refer in any way to the topic of his research. No information related to the use of oil additives. Nothing about effective microorganisms. This section should include a clear research hypothesis.

The Reviewer's comments were included in the corrected article.

Reviewer 3 Report

Review of the manuscript applsci-1840769 with title: "Effect of effective microorganisms, colloidal nanosilver and silver compounds addition on the water content in new and used engine oil", by Krakowski.

This manuscript presents an experimental investigation on the effect of microorganisms, colloidal and soluble silver on the water content of engine oil. In its present form, the overall quality of the manuscript is low. However, I think the topic is interesting and encourage the author to make a good revision of his manuscript. Some major points are given below.

1. The abstract needs extensive revision. Instead of describing the structure of the paper, information should be given about the importance of the application, the research approach and the key findings. Please follow the instructions to the authors provided by the journal.

2. The introduction and discussion are very poorly connected to relevant literature.

3. Critical information is missing from the research methodology about the type of microorganisms, the culture conditions, their concentration in liquid medium, the production of ceramic tubes, all the same for colloidal silver too, the measurement of the flash point and viscosity.

4. The rationale of the research design is not clear. In most cases, additives with concentration of 2.5% and 5% v/v resulted in unmeasurable water content. Perhaps, experiments with smaller amounts (e.g., 0.1 to 2%v/v) should be carried out. The turbidity or droplet size distribution should be measured whenever emulsions form.

5. The reference list contains many web pages [#24-31], which might not be available after a while. Whenever possible, they should be replaced by peer-reviewed research publications. Also, several references [#7,12,15,16,23] are in Polish and limit readers without relevant fluency from verifying the claims of the author and any supporting information.

Round 2

Reviewer 2 Report

Many suggestions from the previous review of the paper were not taken into account by the Author. The new information in the text indicates that the study scheme was not complete or still not correctly described. Perhaps Author needs more time to improve the manuscript.

- Abstract has not been rewritten. The theoretical introduction takes up as much space as the description of the results and conclusions.

- Technically - several consecutive references are not e.g. 1,2,3,4 but 1-4. This relates to the whole manuscript.

- EM information has no sources given. No results or scientific references confirming such properties of effective microorganisms.

- "Lubricating oil...." is part of the introduction and is still a subsection.

- In the chapter Materials and methods, subchapters should be used.

The Materials chapter should be limited to the study-relevant characteristics of the oil, EM (composition, manufacturer) and other additives used. Other information (references) can be placed in the Introduction. Figures 1 and 3 should be removed - they do not contribute anything to the paper.

- No methodology for analysis of dynamic viscosity, flash point.

- Still nothing known about repetition number. Still no basic statistics.

- No information regarding method of microbial colonisation on ceramic tubes.

- What was the amount of microorganisms introduced into the oil in the study (number/mL)?

- Are all the data in the tables results from previous studies? Definitely this refers to the acid number and base number analysis, but are the other results also? There is the discussion chapter for the comparison, and this chapter is still missing in the paper.

- Descriptions on the Y-axis of the figures should be standardised (units) and corrected.

- The last chapter has been completed, but it is now a re-description of the research results rather than the conclusions of the paper.

- Author has written in the results and in the conclusions that the microorganisms did or did not affect the analysed parameters. Although some information was completed, there is no data in the paper on analysing the numbers of microorganisms 4 weeks after treatment with oils. Some microorganisms use oil products as a source of carbon and energy and can grow in such products, but not all of them. Were there such microorganisms in the introduced product? How does Author know that microorganisms grew in the tested oils and changed the parameters of the oils if he did not carry out such analyses? In the study scheme there is no control for the object with microorganisms. If Author has such results they should be included in the paper. If not such conclusions are not justified and this part of the study regarding microorganisms, due to inappropriate methodology, cannot be part of this paper.

- References still need to be improved as required by the Editor.

Author Response

Review of the revised version of the manuscript applsci-1840769 with title: "Effect of effective microorganisms, colloidal nanosilver and silver compounds addition on the water content in new and used engine oil".

- Abstract has not been rewritten. The theoretical introduction takes up as much space as the description of the results and conclusions.

Abstract has been rewritten and the theoretical introduction has been shortened.

- Technically - several consecutive references are not e.g. 1,2,3,4 but 1-4. This relates to the whole manuscript.

It has beed corrected.

- EM information has no sources given. No results or scientific references confirming such properties of effective microorganisms.

Scientific references confirming the properties of effective microorganisms have been added to the article:

  1. Higa, T. Effective microorganisms - technology of the 21st century. Conference "Effective Microorganisms in the World" University of East London, Great Britain, 23 July 2005.
  2. Higa T. A revolution in protecting our planet. Foundation for Development of the Warsaw University of Life Sciences, Warsaw 2003. (In Polish)
  3. Kolasa-Więcek A., Will effective microorganisms revolutionize the world? Advances in Food Processing Technology 2010, 1, 66 – 69. Available online: http://yadda.icm.edu.pl/baztech/element/bwmeta1.element.baztech-article-BPL2-0021-0030 (accessed on 30 August 2022). (In Polish)
  4. Mayer, J.; Scheid, S.; Widmer, F.; Fließbach, A.; Oberholzer, H-R. How effective are ‘Effective microorganisms® (EM)’? Results from a field study in temperate climate. Applied Soil Ecology 2010, 46(2), 230-239. Available online: https://doi.org/10.1016/j.apsoil.2010.08.007 (accessed on 30 August 2022).

- "Lubricating oil...." is part of the introduction and is still a subsection.

Chapter 2 has been renamed. Chapter 2 has been shortened to focus primarily on contamination of the oil with water.

- In the chapter Materials and methods, subchapters should be used.

It has beed corrected.

The Materials chapter should be limited to the study-relevant characteristics of the oil, EM (composition, manufacturer) and other additives used. Other information (references) can be placed in the Introduction. Figures 1 and 3 should be removed - they do not contribute anything to the paper.

The contents of the Materials chapter have been limited to the minimum necessary. Figures 1 and 3 should be removed.

- No methodology for analysis of dynamic viscosity, flash point.

Dynamic viscosity and flash point have been referred to in order to confirm the results obtained and are the subject of other articles, one of which (flash point) is under development:

Krakowski, R. Research into the effects of the effective microorganisms addition on the engine oil viscosity Journal of KONES 2019, 26 (3), 105-112. Available online: https://kones.eu/ep/2019/vol26/no3/105-112_JOK_2019_NO.3_VOL.26_ISSN_1231-4005-KRAKOWSKI.pdf

- Still nothing known about repetition number. Still no basic statistics.

Each sample of pure oil and oil with additives were tested three times. However, the article presents the average result for each case. This information was added to Chapter 4 "Results and analysis of research"

No information regarding method of microbial colonisation on ceramic tubes.

Effective microorganisms in the form of ceramic tubes are a special type of clay inoculated with beneficial microflora (fermented clay), which mature in natural conditions for several months, then it is fired under high temperatures in anaerobic conditions, compositions of beneficial microorganisms, sugar cane molasses, restructured water. Clays inoculated with microorganisms mature for many weeks in appropriate humidity and temperature, and then, depending on their intended use, they are fired at various temperatures from 500 oC to 1200 oC. The shapes of the fired elements or dishes are adjusted to their future use.

- What was the amount of microorganisms introduced into the oil in the study (number/mL)?

A new and used oil with effective microorganisms in liquid form (2.5 ml and 5 ml per 100 ml of oil) and in the form of ceramic tubes (3 pieces and 6 pieces with a diameter of 9 mm and a height of 11 mm for 100 ml of oil) was used for the research. In addition, silver solution and colloidal nanosilver were added to fresh and used oil in the same proportions as for effective microorganisms (2.5 ml and 5 ml per 100 ml of oil).

I do not know the amount of effective microorganisms in ml of the solution and in a ceramic tube, because there is no such information provided by the producers of these products.

- Are all the data in the tables results from previous studies? Definitely this refers to the acid number and base number analysis, but are the other results also? There is the discussion chapter for the comparison, and this chapter is still missing in the paper.

All the data in the tables results are from previous studies  [55-57]. There is such information in the article and a link to the publication. In the article, they are intended to confirm the correctness of the obtained results, and therefore they have not been discussed in greater detail, because they are not the main topic of the article.

- Descriptions on the Y-axis of the figures should be standardised (units) and corrected.

It has beed corrected.

The last chapter has been completed, but it is now a re-description of the research results rather than the conclusions of the paper.

It has beed corrected.

- Author has written in the results and in the conclusions that the microorganisms did or did not affect the analysed parameters. Although some information was completed, there is no data in the paper on analysing the numbers of microorganisms 4 weeks after treatment with oils. Some microorganisms use oil products as a source of carbon and energy and can grow in such products, but not all of them. Were there such microorganisms in the introduced product? How does Author know that microorganisms grew in the tested oils and changed the parameters of the oils if he did not carry out such analyses? In the study scheme there is no control for the object with microorganisms. If Author has such results they should be included in the paper. If not such conclusions are not justified and this part of the study regarding microorganisms, due to inappropriate methodology, cannot be part of this paper.

„…there is no data in the paper on analysing the numbers of microorganisms 4 weeks after treatment with oils” - I do not know and I have not tested the number of effective microorganisms after 4 weeks.

….Were there such microorganisms in the introduced product? New and used oils were used in the research. There may be microorganisms in the used oil that use petroleum products for development, but I don't know exactly because it has not been studied.

Gaylarde C. C., Bento F., Kelley J., Microbial contamination of stored hydrocarbon fuels and its control, Revista de Microbiologia, nr 30, str. 1 -10, 1999

Hill E. C., Biodegradation of petroleum products. Petroleum microbiology, red. R. M. Atlas, Macmillan Publ, London, 1984.

Janda, K. Microbiological contamination of fuels. Advances in microbiology 2005, 44, 157-169. (In Polish).

How does Author know that microorganisms grew in the tested oils and changed the parameters of the oils if he did not carry out such analyses? I do not write in the article that the microorganisms grew in the tested oils and that it influenced the parameters in the oils, I added two different amounts, i.e. 2.5 and 5 ml of the solution and checked how it affects the parameters of the oils.

- References still need to be improved as required by the Editor.

It has beed corrected.

Reviewer 3 Report

Review of the revised version of the manuscript applsci-1840769 with title: "Effect of effective microorganisms, colloidal nanosilver and silver compounds addition on the water content in new and used engine oil", by Krakowski.

This manuscript presents an experimental investigation on the effect of microorganisms, colloidal and soluble silver on the water content of engine oil. The revision was decent at some point, but hasty at others. Specific suggestions that could further improve this paper are given below.

1. Without additional experiments for lower EM/nanosilver concentrations, and without statistical analysis (triplicate experiments), this paper presents a preliminary study, instead of a systematic one. Therefore, I would suggest to change the title as follows: "Effect of effective microorganisms, colloidal nanosilver and silver compounds addition on the water content in new and used engine oil: a preliminary study"

2. The abstract is messy and needs to be tidied up. I would suggest to start by stating the key application, e.g. "Water and microorganisms are major factors of contamination for engine oils and fuels and lead to a significant reduction in the lifetime and performance of engines.",  and remove everything between "At the beginning of the article ... compounds addition were shown".

3. The introduction remains poorly connected to relevant literature. I would suggest to cite the following highly-relevant papers:

- Page 1, first paragraph: "microbial contamination ... is becoming more and more common". At this point, the recent paper by Vidal-Verdú et al [1] on the abundance of oil-degrading microbes at the tank lid of cars, could be cited.

- Page 1, first paragraph: "Such protection ... chemical method". Another method of protection against microbial contamination is the use of specialized coatings on fuel tanks [2].

- Page 2, second paragraph: "Another environmentally friendly measure can be the use of silver". At this point, a seminal paper by  Ramirez et al. [3] on the antibiotic activity of silver, could be cited.

- Page 2, third paragraph: " Good quality petroleum products ... turbidity and darkening". At this point, a very interesting paper by Prof. Okpokwasili [4] on microbial growth in brake fluids could be cited.

- Page 2, fourth paragraph: " The point ... is called the oil saturation point". Here, the extremely useful paper by Fregolente et al [5] that supports the statements on the saturation point should be cited.

4. In the third paragraph, the author claims that "One of the methods of combating microorganisms, ..., may be effective microorganisms (EM)". Is there a reference in support of this statement?

5. In the Results, page 8, it is stated that "These photos show ... was intended." At this point, the author should briefly discuss about possible causes for the "milky appearance" of engine oil when liquid EM and colloidal silver is added. For instance, it could be mentioned that the milky appearance may be the outcome of Pickering emulsification [6-8], in which the bacteria and colloidal silver accumulate at the water-oil interface and prevent the water-in-oil droplets from coalescing. In the case of EM, emulsification may also be caused by biosurfactants [9].

6. According to the provided links, the liquid EM contains molasses, which is rich in sugars and exhibits a shear-thinning rheological behavior. Are there any possible adverse effects of molasses on the composition and rheology of engine oil? Perhaps, add a brief comment in page 9 about the effects on viscosity.

Minor (editorial) points:
+ The link for Reference 34 (related to figure 1) is not accessible.
+ The image shown in Figure 3 does not exist in Reference 38. The correct source must be cited.
+ In my copy of the manuscript, page 3 is blank. If not intentional, please add whatever information is missing.
+ The copyright policy and any relevant permission to use the images in Figures 1-4 should confirmed.

Suggested references
[1] Vidal-Verdú, À.; Gómez-Martínez, D.; Latorre-Pérez, A.; Peretó, J.; Porcar, M. The car tank lid bacteriome: a reservoir of bacteria with potential in bioremediation of fuel, npj Biofilms Microbiomes 2022, 8, 32. https://doi.org/10.1038/s41522-022-00299-8
[2] Zhang, J.; Luo, H.; Yin, X.; Shi, Y.; Zhang, Y.; Tan, L. Surface coating on aluminum substrate with polymeric guanidine derivative to protect jet fuel tanks from microbial contamination, Surface and Coatings Technology, 2021, 422,127521. https://doi.org/10.1016/j.surfcoat.2021.127521.
[3] Morones-Ramirez, J.R.; Winkler, J.A.; Spina, C.S.; Collins, J.J. Silver enhances antibiotic activity against Gram-negative bacteria, Science Translational Medicine 2013, 5(190), 190ra81. https://doi.org/10.1126/scitranslmed.3006276
[4] Maduka, C.M.; Okpokwasili, G. C. Microbial growth in brake fluids, International Biodeterioration & Biodegradation 2016, 114, 31-38. https://doi.org/10.1016/j.ibiod.2016.05.017.
[5] Fregolente, P.B.L.; Fregolente, L.V.; Wolf-Maciel, M.R. Journal of Chemical & Engineering Data 2012, 57 (6), 1817-1821. https://doi.org/10.1021/je300279c
[6] Binks, B.P. Particles as surfactants – similarities and differences. Curr. Opin. Colloid & Interface Science 2002, 7, 21–41. https://doi.org/10.1016/S1359-0294(02)00008-0
[7] Kapellos, G.E. "Microbial strategies for oil biodegradation" in
Modeling of Microscale Transport in Biological Processes, Academic Press (Cambridge, MA), 2017, 19-39. https://doi.org/10.1016/B978-0-12-804595-4.00002-X
[8] Valdivia-Rivera, S.; Ayora-Talavera, T.; Lizardi-Jiménez, M.A.; García-Cruz, U.; Cuevas-Bernardino, J.C.; Pacheco, N. Encapsulation of microorganisms for bioremediation: Techniques and carriers, Reviews in Environmental Science and Bio/Technology 2021, 20, 815–838. https://doi.org/10.1007/s11157-021-09577-x
[9] Geetha, S.J.; Banat, I.M; Joshi, S.J. Biosurfactants: Production and potential applications in microbial enhanced oil recovery (MEOR), Biocatalysis and Agricultural Biotechnology 2018, 14, 23-32. https://doi.org/10.1016/j.bcab.2018.01.010.

Author Response

Review of the revised version of the manuscript applsci-1840769 with title: "Effect of effective microorganisms, colloidal nanosilver and silver compounds addition on the water content in new and used engine oil", by Krakowski.

  1. Without additional experiments for lower EM/nanosilver concentrations, and without statistical analysis (triplicate experiments), this paper presents a preliminary study, instead of a systematic one. Therefore, I would suggest to change the title as follows: "Effect of effective microorganisms, colloidal nanosilver and silver compounds addition on the water content in new and used engine oil: a preliminary study"

I agree with the Reviewer's suggestion, the title will be changed: "Effect of effective microorganisms, colloidal nanosilver and silver compounds addition on the water content in new and used engine oil: a preliminary study"

  1. The abstract is messy and needs to be tidied up. I would suggest to start by stating the key application, e.g. "Water and microorganisms are major factors of contamination for engine oils and fuels and lead to a significant reduction in the lifetime and performance of engines.",  and remove everything between "At the beginning of the article ... compounds addition were shown".

The abstract has been corrected as suggested by the Reviewer.

  1. The introduction remains poorly connected to relevant literature. I would suggest to cite the following highly-relevant papers:

- Page 1, first paragraph: "microbial contamination ... is becoming more and more common". At this point, the recent paper by Vidal-Verdú et al [1] on the abundance of oil-degrading microbes at the tank lid of cars, could be cited.

The paper has been cited.

Page 1, first paragraph: "Such protection ... chemical method". Another method of protection against microbial contamination is the use of specialized coatings on fuel tanks [2].

The paper has been cited.

- Page 2, second paragraph: "Another environmentally friendly measure can be the use of silver". At this point, a seminal paper by  Ramirez et al. [3] on the antibiotic activity of silver, could be cited.

The paper has been cited.

- Page 2, third paragraph: " Good quality petroleum products ... turbidity and darkening". At this point, a very interesting paper by Prof. Okpokwasili [4] on microbial growth in brake fluids could be cited.

The paper has been cited.

- Page 2, fourth paragraph: " The point ... is called the oil saturation point". Here, the extremely useful paper by Fregolente et al [5] that supports the statements on the saturation point should be cited.

The paper has been cited.

  1. In the third paragraph, the author claims that "One of the methods of combating microorganisms, ..., may be effective microorganisms (EM)". Is there a reference in support of this statement?

An article confirming this statement has been added.

  1. In the Results, page 8, it is stated that "These photos show ... was intended." At this point, the author should briefly discuss about possible causes for the "milky appearance" of engine oil when liquid EM and colloidal silver is added. For instance, it could be mentioned that the milky appearance may be the outcome of Pickering emulsification [6-8], in which the bacteria and colloidal silver accumulate at the water-oil interface and prevent the water-in-oil droplets from coalescing. In the case of EM, emulsification may also be caused by biosurfactants [9].

It has been added to the article.

  1. According to the provided links, the liquid EM contains molasses, which is rich in sugars and exhibits a shear-thinning rheological behavior. Are there any possible adverse effects of molasses on the composition and rheology of engine oil? Perhaps, add a brief comment in page 9 about the effects on viscosity.

It has been added to the article.

Minor (editorial) points:

+The link for Reference 34 (related to figure 1) is not accessible.

Following the suggestion of the second Reviewer, the drawing has been removed.

+ The image shown in Figure 3 does not exist in Reference 38. The correct source must be cited.

Following the suggestion of the second Reviewer, the drawing has been removed.

+ In my copy of the manuscript, page 3 is blank. If not intentional, please add whatever information is missing.

This is not intended. This is probably an editorial error. In my manuscript, page 3 is not blank.

+ The copyright policy and any relevant permission to use the images in Figures 1-4 should confirmed

The photos present the products in a technical way and come from the website of the online store, so they are not creative. For this reason, no special confirmation of permission to use the images is required.
